# Can Homes Affect Well-Being? A Scoping Review among Housing Conditions, Indoor Environmental Quality, and Mental Health Outcomes

**DOI:** 10.3390/ijerph192315975

**Published:** 2022-11-30

**Authors:** Alessia Riva, Andrea Rebecchi, Stefano Capolongo, Marco Gola

**Affiliations:** 1School of Architecture, Urban Planning, Construction Engineering (AUIC) of Politecnico di Milano, 20133 Milano, Italy; 2Design & Health Lab., Department of Architecture, Built Environment and Construction Engineering (ABC) of Politecnico di Milano, 20133 Milano, Italy

**Keywords:** architectural features, housing conditions, indoor environmental quality, mental health, design recommendations

## Abstract

The purpose of the scoping review is to explore the relationship between housing conditions, indoor environmental quality (IEQ), and mental health implications on human well-being. In fact, time spent at home increased due to the recent COVID-19 lockdown period, and social-sanitary emergencies are expected to grow due to the urbanization phenomenon. Thus, the role of the physical environment in which we live, study, and work, has become of crucial importance, as the literature has recently highlighted. This scoping review, conducted on the electronic database Scopus, led to the identification of 366 articles. This, after the screening processes based on the inclusion criteria, led to the final inclusion of 31 papers related specifically to the OECD area. The review allowed the identification of five housing conditions [house type, age, and floor level; housing qualities; household composition; neighborhood; green spaces] that, by influencing the IEQ parameters, had impacts on the mental health outcomes addressed. By synthesizing the contributions of the review, a list of design recommendations has been provided. These will serve as a basis for future researchers, from which to develop measures to reduce inequalities in housing by making them healthier, more resilient, and salutogenic.

## 1. Introduction

The World Health Organization (WHO) declared the COVID-19 outbreak a public health emergency of international concern on the 30 January 2020 because it has spread to such an extent worldwide, affecting a great number of people [1]. The restricted measures of societal lockdowns, which almost every country put in place, undoubtedly reduced the impact of the pandemic [2]. At the same time, they have been detrimental to the quality of life, both physical and mental. Indeed, throughout the pandemic and related lockdowns, time spent at home increased to a great extent [3].

Prior to the COVID-19 pandemic, the population spent 60% of their time at home [2]. Now it is well-known that humans spend over 85% of their time in indoor settings. In fact, the COVID-19 transformed the daily lives of millions of people [3] during its worst outbreaks. It also left permanent changes in our way of living, working, studying, interacting, etc. [4,5].

Furthermore, it is well-established that 55% of the population lives in urban versus rural environments, a trend that has been increased by 30% since 1950 [6]. If urbanization continues this way, as expected, it is estimated that by 2050 around 70% of the population will live in urban environments [6]. In addition to the positive benefits of living in urban areas (such as greater access to healthcare, improved educational opportunities, higher wages, etc.) [6], the time spent indoor will increase. Indeed, urban areas usually lack opportunities to have large amounts of contact with nature, compared to rural ones [2].

Therefore, since in the immediate future we are likely to continue spending more time in confined spaces, the role of the built environments in which we live has become of the utmost importance, as the pandemic has strongly highlighted [7]. A rapidly growing literature indeed has recently investigated the effect of the COVID-19 pandemic and lockdowns on the population’s health [3], noticing a worsening of non-communicable diseases (chronic diseases such as diabetes and mental disorders such as anxiety [2]).

Thus, housing plays a key role in human lives, impacting education, employment, recreation, social opportunities, and most of all health [8]. The WHO describes the physical environment as one of the three determinants of health, in fact, “*many factors combine together to affect the health of the individuals and communities*” [9] along with the social and economic environment and the individual characteristics and behaviors [10,11]. 

Many studies [12,13,14,15,16] have already established a relationship between the built environment and its effects on physical health. Its impact on mental health and well-being however, is a recent issue which is still lacking assessment with quantitative data. The data available are indeed mostly qualitative and mainly referred to developed countries. These huge gaps in the existing literature highlighted the need to start a systematization work that will serve as a basis for future developments.

The purpose of this paper is therefore to synthesize the existing literature on the impact of the built environment in the residential setting—housing conditions—on mental health. Since architects and engineers have historically reimagined and redesigned buildings and cities to face societal changes occurred throughout history [7], this paper will provide a collection of design suggestions that have emerged from the literature. These will help future researchers to have a starting point from which to develop measures to reduce inequalities in housing by making them healthier, more resilient, and salutogenic.

The paper has been structured around the relationship between three domains [indoor environmental quality, housing conditions, mental health] that continuously interact with and influence each other, making it difficult to indicate a univocal direction between them. Specifically, the authors intended them as a direct pathway that, by starting from the housing conditions (WHERE), reached both the indoor environmental quality parameters (WHAT) and mental health (OUTCOMES). 

Additional information, when available, such as socio-economic status, will guarantee a more exhaustive reading of the evidence. The three domains are described as follows: 1.1Indoor Environmental Quality (IEQ)1.2Housing Conditions1.3Mental Health

### 1.1. Indoor Environmental Quality (IEQ)

The built environment has been defined in many ways such as “*man-made structures, features and facilities viewed collectively as an environment in which people live and work*” [17], and it generally encompasses all the aspects of our surroundings, from the buildings in which we live in to the distribution and transportation systems [18]. In order to follow the purpose of this paper, the authors limited the definition of the built environment to its factors that are directly correlated with the housing conditions, thus, focusing on IEQ. 

It is defined as a building’s indoor environment performance [19] i.e., the quality of a confined setting in relation to the health and well-being of those who occupy space within it [20]. It encompasses diverse sub-domains (or parameters) that affect human life inside a building [21]. Since there is not a univocal classification of them in the existent literature, we considered the following five IEQ parameters and referred to them, as most literature reviews did. A brief explanation is reported below, along with any other terms we found among the scientific literature referred to them. They are:**Indoor Air Quality (IAQ):** In general, it depends on airborne contaminants inside a, i.e., building, pollutants, malodorous irritants, etc. [21]. According to WHO, air pollution is a leading environmental risk to health and a major contributor to the burden of disease worldwide [22], with more than 80% of the urban population being exposed to air quality levels that exceed the health-based guidance values [23]. Air pollution research was concentrated on outdoor air until radon and formaldehyde health concerns emerged in 1960s and 1970s [6];**Thermal comfort:** It includes air temperature, air velocity, radiant temperature, humidity, and relative humidity. It is acceptable if at least 80% of the occupants feel comfortable [21]; it can be provided by natural or mechanical ventilation;**Lighting comfort:** The main aspects are light level (intensity or brightness), contrast, and glare [21]; it is also well-known as visual comfort or daylight;**Acoustic comfort:** This refers to the quality of sounds inside a building [21]; it is also called aural comfort or noise level;**Overcrowding:** Amount of space, visual privacy, and ease of interaction. According to WHO, living space must be such as to “*[…] guarantee adequate privacy in order to meet the needs of the occupants, be accessible and usable for extended users and be large enough to comfortably accommodate people of different ages*”, as Appolloni et al. have stated [24].

### 1.2. Housing Conditions

The authors focused the attention on residential settings by studying which factors characterized them most. From the existing literature, the authors found the relevance of the following five sub-domains [25,26,27]:**House type, age, and floor levels:** Such as the apartment/flat layout, shared houses, typology of rooms, condominiums, flats, detached houses, their construction year, floor levels, etc.;**Housing qualities:** Such as number of bedrooms, type of apertures, spaces’ dimensioning, the presence of structural problems, type of air system, problems of condensation, energy efficiency retrofits, etc.;**Household composition:** Such as family members, number of people at home, etc.;**Neighborhood:** Such as urban or rural areas in which the building is localized, presence of infrastructures, condition of pollution, etc. [28];**Green spaces:** Such as green spaces, viewing nature, exposure to nature, gardening in the home, etc. [29,30].

### 1.3. Mental Health

According to WHO, mental health is “*a state of mental well-being that enables people to cope with the stresses of life, realize their abilities, learn well and work well, and contribute to their community*” [31]. It includes users’ emotional, psychological, cognitive, behavioral, and social well-being, affecting how they think, feel, and act [32]. 

The quality of mental health can be protected or undermined throughout our lives by modifiable (such as socio-economic conditions, employment, education, social involvement, and housing quality) and unmodifiable factors (such as gender, age, ethnicity, etc.) [33].

Therefore, mental health is “*[…] more than just the absence of mental disorder or disabilities*” [31], and, as stressed out by WHO, “*mental health conditions include mental disorders and psychosocial disabilities as well as other mental states associated with significant distress, impairment in functioning, or risk of self-harm*” [31]. 

The prevalence of mental health conditions is increasing worldwide in recent decades, leading to a growing acknowledgment of the important role that they play in achieving global development goals [34]. This is testified by the inclusion of mental health in the Sustainable Development Goals (SDG) [34]. Furthermore, the COVID-19 pandemic had a great influence on these conditions, and its effects will be demonstrated in the upcoming years [35].

There are more than 200 types of mental disorders/illnesses [36,37]. Focusing on developed countries, they are estimated to affect one in five adults [36], one in five children [36], and one in six youth aged 6–17 [38] in any given year, causing one in five years lived with disability [34]. It is also estimated that in the US more than 50% of people will be diagnosed with a mental illness at some point in their lifetime and that 4.5% (11.2 million) of adults had a severe psychological condition in 2017 [33]. Additionally, the lost earning for mental health illness in the US in 2016 amounted to 193.2 billion US dollars [6]. Quantitative information about the low–middle income countries is still not exhaustive, thus making it impossible to use them as a comparison source. 

For the purpose of the paper, the authors also considered overall mental health and the following three mental health disorders as outcomes: **Depression:** It is characterized by depressive mood or a loss of pleasure or interest in activities for most of the day, nearly every day, for at least two weeks. Other symptoms are poor concentration, feelings of excessive guilt or low self-worth, hopelessness about the future, thoughts about death or suicide, disrupted sleep, changes in appetite or weight, and feeling overly tired or low in energy. It is estimated that it affected 280 million people in 2019, including 23 million of children and adolescents [39];**Anxiety:** It is characterized by excessive fear, worry and related behavioral disturbances. There are several kinds of anxiety disorders: generalized anxiety disorder, panic disorder, social anxiety disorder, and separation anxiety disorder. It is estimated that it affected 301 million people in 2019, including 58 million children and adolescents [40];**Stress:** It is defined as the feeling of being under pressure or threatened. It is classified as acute or chronic, which can lead to anxiety or depression [41].

The authors reported six additional conditions as mental health outcomes. Although they are not strictly considered to be mental disorders, they indicate a state of well-being correlated with mental health illnesses, either depending on or causing them. They are:sleep problems [42];loneliness;cognitive fatigue;positive feelings about life satisfaction, self-esteem, and motivation;negative feelings about irritability, aggression, and frustration;productivity.

## 2. Materials and Methods

The scoping review was conducted following the method designed by the “*Johanna Briggs Institute methodology for scoping review*” [43,44]. In particular, the authors referred to the search framework proposed by Arksey and O’Malley [45], which is composed of the following stages, described from Section 2.1, Section 2.2, Section 2.3, Section 2.4, Section 2.5 and Section 2.6.

### 2.1. Definition of the Research Questions

The main research questions, promoter of the review, are: What is the impact of Housing Conditions on Mental Health and which IEQ parameters influence Mental Health outcomes in residential settings?

### 2.2. Search Strategy

The search strategy for retrieving the studies was carried out through the electronic database Scopus and was developed on 8 June 2022. It consisted in a range of keywords coherent with the main topic and the research question, logically combined using Boolean operators. Each of the three domains described in the introduction [IEQ, housing conditions, mental health] were used as keywords. However, in order to obtain wider and more likely results, each of them has been further sub-categorized with several keywords, as Figure 1 shows.

In fact, from a preliminary hand search review, the authors noticed that a comprehensive literature regarding these topics is still missing, making it impossible to univocally refer to one of them. Thus, the authors also used synonyms and various parameters by which these areas are characterized as keywords. 

The search strings obtained and shown in Figure 2 led to the identification of a total of 1052 contributions.

### 2.3. Inclusion and Exclusion Criteria

In order to proceed with the screening process, the authors defined the inclusion criteria according to the research question, as Figure 3 shows. We reviewed only original papers and peer-reviewed studies in order to make our research as valid as possible. Other non-original studies such as dissertations, conference proceedings, editorials, comments, grey literature, book chapters, correspondence, and brief notes were excluded. Similarly, studies afferent to other disciplinary fields, whose outcomes were not strictly related to the paper’s scope (such as dentistry, nursing, chemistry, veterinary, mathematics, etc.), were excluded.

In addition, to improve internal validity, the authors set a geographic limit, by including only studies conducted in the Organization for Economic Co-operation and Development (OECD) area. That is because most of the information available was about developed countries. The huge gap between data and studies regarding housing conditions and mental health in under-developed countries made it impossible to include them in this article. Thus, low–middle-income countries were excluded on purpose. 

Furthermore, as recommended by the Cochrane collaboration [46], the authors also set language and time filters. Thus, only studies that were published in the English language from 2012 to the present day were included. This data range was chosen because the authors noticed that only in recent years has there been a great deal of interest in these topics, resulting in more data available. 

The authors accepted both self-reported measures and data extracted from clinical databases, as well those as self-assessed by interviews for mental health outcomes. Studies with analytical purposes and a design and operational approach were included. Both prospective and retrospective cohort studies were eligible for inclusion. 

The authors selected articles regarding the general population, without any limitations to age and gender. Studies related to the pandemic period were also included, since this is an issue of upcoming interest, and it has been proven to have strong impacts on the topics of this review paper. 

Studies that examined homelessness and specific and chronic mental illnesses (such as autism, dementia, developmental disorders, schizophrenia, etc.) were excluded on purpose since they referred to specific clinical situations that are not strictly related to housing conditions. 

The search finally resulted in 366 articles.

### 2.4. Study Selection

According to the preferred reporting items for systematic reviews and meta-analyses (PRISMA) [47,48] flow diagram, as Figure 4 shows, the authors reported the double-pass screening process carried out by three reviewers who independently analyzed all the 366 identified records in order to reduce any individual bias. No duplicates were found.

The first step was the screening of titles, author keywords, and index keywords to assess potential eligibility. This led to 62 papers that were further checked by reading the abstracts, leading to 41 articles whose abstracts contained relevant data. The second step checked the articles by reading through the full texts. As a result, a total of 31 articles were finally included as they met the inclusion criteria. In fact, the analysis of the full texts led to the exclusion of 10 studies, as their outcomes did not fully satisfy the research question addressed. In particular:*n* = 2 were excluded because they referred to undergraduate students in college dorms [49] and children in daycare centers [50]. Thus, they were not coherent with the residential settings;also, a study by Srinivasan and Ram [51] was excluded because it was preliminary research that still lacks any results (*n* = 1);in addition, although it was related to the scope of the research, the study by Grazuleviciene et al. [52] was highly focused on blood pressure (*n* = 1);instead, Singh [53] did not consider any of the IEQ parameters nor any housing conditions (*n* = 1);in conclusion, other studies [54,55,56,57,58] were excluded because the authors did not find the full texts (*n* = 5).

### 2.5. Data Extraction

For each included study, relevant data were extracted, compiled, and plotted in a grid developed by the authors and based on the Joanna Briggs Institute data charting model [43,44]. The grid was organized into sections:Section 1 and Section 2 provided general information about the study and the participants involved, respectively;Section 3 provided information about the socio-economic status of the participants;Section 4, Section 5 and Section 6 concern, respectively, the three domains described in the Introduction;finally, the two last columns were about the COVID-19 pandemic and eventual strategies of policies suggested in the papers themselves.

Both qualitative and quantitative data were extracted. When the same study examined associations between multiple forms of Housing Conditions and Mental Health outcomes, data were extracted separately for each of the associations. Each section is summarized in Figure 5.

### 2.6. Brief Sum Up on the Evidences

From the grid developed by the authors, the following preliminary clusterization emerged, as Figure 4 shows. The 31 studies analyzed have been sub-divided into:regarding the typology of paper:-*n* = 5 were position papers;-*n* = 6 were review papers;-*n* = 20 original papers.regarding the geographical localization of the studies conducted:-*n* = 3 studies were conducted in the European Union (EU),-*n* = 5 in South America (SA),-*n*= 6 in the US (North America),-*n* = 3 in Eastern Countries (ME),-and in *n* = 14 the country was not specified.regarding the five IEQ parameters investigated, the authors observed:-*n* = 15 contributions referred to IAQ;-*n* = 18 studies referred to thermal comfort;-*n* = 8 to lighting comfort;-*n* = 8 to acoustic comfort;-*n* = 11 referred to overcrowding.regarding the five Housing Conditions, the authors observed:-*n* = 13 referred to house type, age, and floor levels;-*n* = 17 referred to housing qualities;-*n* = 10 to household composition;-*n* = 12 to neighborhood;-*n* = 8 referred to green spaces.regarding Mental Health outcomes, the authors observed:-*n* = 27 studies considered the overall mental health;-*n* = 24 addressed the depression outcomes;-*n* = 15 addressed the anxiety outcomes;-*n* = 24 addressed the stress outcomes;-*n* = 8 referred to sleep problems;-*n* = 5 referred to loneliness;-*n* = 11 referred to cognitive fatigue;-*n* = 7 referred to positive feelings;-*n* = 9 referred to negative feelings;-*n* = 4 referred to productivity. 

In general, none of the studies referred to the full set of IEQ parameters. *n* = 6 studies were about the COVID-19 pandemic and its consequences. In addition, overall, most of the studies (*n* = 16) addressed the general population without any limitations to gender and age. *n* = 1 specifically addressed university students. *n* = 3 studies specifically addressed children and young adults and the remaining, *n* = 11 studies did not provide any specifications. 

It must be noticed that the boxes of the grid filled with “*cited but without any specific information*” are not considered in this paragraph.

In order to synthetize the contents covered by the selected studies, we used VOSViewer for performing co-occurrences analyzes on terms from titles and abstracts. This way, we were able to gain an immediate visualization of the main topics of the selected studies. In particular, the “*Overlay Visualization*” shown in Figure 6 allowed the subdivision of the publications and their related terms based on the average publication year of the documents in which a keyword occurs, or the average publication year of the documents published by a source, author, organization, or country.

## 3. Results

The grid developed by the authors was fundamental to the results reading, which tried to analyze the relationship between each subcategory of the housing conditions section and each subcategory of the IEQ and mental health sections, as schematically explained in Figure 7. Consequently, we were able to recognize the factors that had the greatest impact on each subcategory. 

Similarly, we were able to highlight the existing relationships between these factors and point out whether and where these relationships were missing. Furthermore, in case data were available, the authors reported both negative and positive responses for some outcomes, since sometimes one of them did not certainly exclude the other. 

### 3.1. House Type, Age, and Floor Level

These housing features were investigated in 13 studies. On average, these articles referred to apartments [3,59], houses [3,59], flats [60], and multi-unit residential buildings [61,62]. One study specifically referred to shared houses [63], one to luxury apartments [64], and three to high-rise buildings [6,65,66]. Four studies referred to detached houses [60,67,68,69]. Among the 13 studies, the construction year and the floor level were taken into account only in three studies [64,67,68]. 

Across the IEQ parameters, associations were found with IAQ [67] and acoustic comfort [60,61]. Regarding the IAQ, the study by Rickenbacker et al. [67] showed that radon concentration levels were found to be 69% higher in homes built before 1940 than homes built after the same date. Additionally, they were found to be higher in lower floors since radon seeps directly from foundations and enters homes through basements and crawlspaces [70].

Regarding acoustic comfort, Torresin et al. [60] and Andargie et al. [61] found a negative association between flats compared to detached houses, since respondents who have neighbors are more likely to experience worse acoustic comfort due to the noise produced. Lastly, Andargie et al. [61] found that noise annoyance was higher in older than newer buildings and among respondents in lower floors.

Across the mental health outcomes, associations were found with overall mental health [6,60,65,66,67], depression [6,63], anxiety [6,63], stress [6,61,65] and loneliness [65]. Overall mental health was found to be worse due to living in high-rise housing [6,66], particularly among women and children [65], as it limited social interactions and play opportunities.

Other predictors were living in apartment complexes [67], which led to social isolation because of the lack of common spaces and opportunities for social interactions, and due to living on higher floors [6,65], which also restricted social interactions.

Furthermore, the presence of neighbors [60] and their related noise in non-isolated houses caused a worsening in overall mental health. When assessing depression and anxiety, a negative association was found with both factors, as well as living in shared housing [63] and in high-rise housing because of the limited opportunities for social interactions [6]. Across the two studies focusing on the stress outcomes, the main predictors of its higher levels were living in high-rise housing, which causes the limited social interactions [6], in particular among women and children [65], and the presence of neighbors and related noise [61]. Living in high-rise building was also correlated with higher levels of loneliness in women [65].

Lastly, a statistically significant association was not found between overall mental health and dwelling types [3]. Similarly, anxiety, loneliness, and positive feelings were never found to be correlated with living in apartments compared to houses [3].

### 3.2. Housing Qualities

These features were examined in 17 studies. Among the included studies, two [59,61] referred to the number of available bedrooms, three [2,60,68] to the house size, and another by Torresin et al. [60] referred to the availability of a quiet side in the house. The number of windows, their proportions, size, and openings’ frequency were examined in four studies [7,61,67,71]. Other features addressed were the type of air system [60,67,69], the presence of unpleasant odors [6], the proximity to elevators and garbage chute [61], the presence or absence of balconies and terraces [2,61], as well as the basement and fuel use [67].

On average, most of the studies referring to housing inadequacy intended it to signify structural problems [2,8,65,66,72,73], presence of damp, mold, and condensation [2,72], maintenance and upkeep [65,73], amenities [65], leaking roofs [72], lack of adequate heating [72], cleanliness, and clutter [66,73]. Another cluster of studies referred to energy conservation retrofits such as insulation [6,61,62,72,74], double glazing [74], new boilers and kitchens [72], and central heating [62,74]. 

Regarding IEQ parameters, associations were found with IAQ [6,7,65,67,69,72,74], thermal comfort [7,69,72], lighting comfort [71], acoustic comfort [6,60,61] and overcrowding [60].

Concerning IAQ, some studies showed that the type of air system can influence air quality. An inadequate ventilation system increases the indoor levels of carbon dioxide [67], and overall levels of air pollutants concentrations [6,74], and alters the exposure to ambient pollutants [65]. When correctly managed though, as Bernal demonstrated, mechanical ventilation can help to reduce the presence of pollutants [69].

Natural ventilation—through frequent window and door openings—contributes to creating a deposition of dust particles but it is helpful to dissipate carbon dioxide and prevent its accumulation [7]. The study by Poortinga et al. examined how energy efficiency retrofits, such as the ones listed in the previous paragraph, can help to reduce exposure to pollutants, allergenic spores, mold growth, and damp [72].

Damp and mold growths were also found to be reduced by lower indoor temperatures [72]. When addressing the thermal comfort parameter, it was found to be improved by frequent air exchanges through operable windows and doors [7], as well as by energy efficiency retrofits concerning heating and insulation improvements [72]. 

The lighting comfort parameter was examined only in one study [71] that showed how the number and proportions of apertures in homes can alter daylight penetration.

Differently, the Acoustic Comfort was positively correlated with the house size [60], with a higher level of comfort being registered in homes larger than 80 square meters. Additionally, positive correlations were found with the space availability in the homes [60], the type of air system [60], and the presence of extra noise insulation and paned windows [6]. Negative associations were due to the proximity to elevators and garbage chutes [61] in addition to the airborne and structure-borne noise from HVAC systems [61]. Finally, one study [59] showed a relationship with overcrowding, linking it to the number of available bedrooms while also considering the number of persons present in the household and the house size.

Regarding mental health outcomes, associations were found with:overall mental health [6,8,65,66,72,74];depression [6,59,65,66,68,71,74];anxiety [6,66];stress [7,62,65,66,68,71,72,74];sleep problems [7,61];positive feelings [8];negative feelings [66];and productivity [2,71].

Overall mental health was found to be worse when occupants are exposed to lower housing quality [6,65,66] (intended as moisture damage, cleanliness, hazard, and privacy). Other predictors were:living in cold and damp housing [8,72,74];the presence of cavity wall insulation [72];and when occupants are affected by changes in housing conditions among low-income women [66].

Overall mental health was found to be better when there were more windows in a room, guaranteeing the right amount of daylight exposure and a pleasant lighting quality [65,71], and when having adequate space in the home [2]. Additionally, housing improvements [72] (such as new kitchens, bathrooms, and electrics,) and household energy efficiency interventions [62,74] were registered as increasing the overall mental health by reducing household energy expenditures. When addressing the depression outcomes, negative impacts on it were produced by inadequate or non-functioning houses [59], with women reporting 0.8 more depressive symptoms than men on average.

Other predictors were changes in the number of bedrooms available by increasing the household density [59], the inadequate number or lack of windows causing insufficient exposure to daylight [6,65,71], and the presence of dampness and mold [68,74]. Moreover, children growing up in poorer compared to higher-quality housing [66,75] were more likely to experience depressive symptoms. Instead, positive impacts were produced by an adequate exposure through windows to bright, full-spectrum circadian light in the morning between 8 a.m. and 12 p.m. [6,7].

Anxiety was examined in two studies that correlated a lack of windows [6], as well as growing up in poorer compared to a higher quality of housing for children [66], with higher levels of anxiety. With regard to stress, it was negatively correlated with adequate exposure to daylight guaranteed by windows [7,71], availability of space for quiet contemplation, and meditation, and social aggregation [7]. Interventions to reduce damp and mold [74], along with energy and weatherization conservation measures, also had positive impacts [62,68,76].

Positive correlations were found between stress, bright light and glare due to windows, foul odors, and maze-like designs [7]. Poor-quality housing [65] had negative impacts on stress, especially among adults living in lower-quality neighborhoods [66] and in cold and damp houses [72].

When addressing the sleep problems outcomes, one study [7] showed that they decreased when exposed to natural lighting through windows and bright full spectrum circadian sunlight between 8 a.m. and 12 p.m. The study by Engineer et al. [7] also showed that they increased when exposed to light at night and also, depending on the heating and cooling system and their related noise [61].

Analyzing feelings, the positive ones were diminished by living in substandard housing [8]. The negative ones (such as aggressivity) were incremented by lacking adequate spaces in the home [2] and by growing up in poorer quality housing when compared to higher quality housing in children [66].

Lastly, lower levels of productivity were correlated with the absence of adequate spaces in terms of size and flexibility [2], while a higher level of productivity was registered when windows guaranteed adequate visual comfort [71].

### 3.3. Household Compositions

This feature was addressed in 10 studies. On average, all the included studies referred to the number of people present in the household. Some studies specifically pointed out the presence or absence of children [3,59,60,77], whether the person was living alone [3,77], and the exact household composition [67,69].

Regarding IEQ parameters, associations were found with IAQ [67], acoustic comfort [60] and overcrowding [3,59,60,63,65,77,78].

The number of persons present in the household was correlated with the concentration levels of carbon dioxide and ozone [67] since those chemicals are human by-products. Therefore, they increased as the human activities carried out in the house increased. 

Moreover, Torresin et al. [60] showed that acoustic comfort was found to be higher when fewer people were at home.

The remaining studies examined the relationship between household composition and overcrowding. Although they globally correlated an increase in the number of occupants with overcrowding, we found some specifications that needed to be explained. 

Across these studies indeed, Torresin et al. [60] and Holmgren et al. [78] defined overcrowding as the number of people present in the dwelling. Ruiz-Tangle and Urria [59] and Evans [65] defined it as the number of persons per bedrooms (specifically, 2,5 persons was the minimum to consider a dwelling overcrowded [59]). Duarte and Jimenéz-Molina [77] defined it in reverse as the number of bedrooms per person. Lastly, Keller et al. [3] and Raynor et al. [63] defined it as having access to sufficient space and autonomy (specifically, at least 43 square meters per person [3]). 

Regarding mental health outcomes, associations were found with overall mental health [3,59,60,65], depression [59,60,63,77], anxiety [3,77], stress [59,63,65,77], loneliness [3], cognitive fatigue [59,63], positive feelings [3], and negative feelings [59,63]. It must be noticed that, as justified by the previous paragraph, we used here the IEQ parameter of overcrowding and the housing condition of household composition interchangeably. 

Overall mental health was found to worsen due to overcrowded household [3,60,65] especially because of the lack of privacy and control, insufficient personal space, hindering of social interactions, and creating conditions of social repetitiveness [59].

Other predictors were living alone [3] and living in less dense environments in adults ≥25 [3] intended to have access to more square meters per person. We also found positive associations with better mental health and living in dwellings offering more rooms for a given household [3].

Young adults <25 [3] living in under-occupied dwellings [3], in less dense environments were defined as having access to more square meters per person, and the presence of a lower number of people at home also had positive impacts [60].

When addressing depression outcomes, negative impacts were linked to overcrowding in two studies [60,63]. Ruiz-Tagle and Urria [59] linked depression to an increase in household overcrowding/density levels. Duarte and Jiménez-Molina [77] correlated the illness with living in a small household, with the presence of children in the household in younger age ≤35 and with women aged 36–59.

Anxiety outcomes were found to be positively related to low household density amongst young people [3] and living in under-occupied dwellings in adult men and women [3]. It was negatively related to living in under-occupied dwellings in the young [3], living with children amongst women but not men or young [3], living in small household, and having children in the household in younger age ≤ 35 and in women aged 36–59 [77]. The impacts on the stress outcome were analyzed in four studies, all of which pointed out how negative the impact of overcrowding was on stress [59,63,65,77].

Loneliness was addressed just by Keller et al. [3] There was a positive correlation among young people for living in under-occupied dwellings and in crowded dwellings. He also pointed out a negative correlation between loneliness, living alone and living in crowded dwellings in adult men and women.

The studies by Ruiz-Tangle and Urria [59] and Raynor et al. [63] deepened the association between the negative impacts of overcrowding and cognitive fatigue. The latter worsens in these conditions due to the lack of opportunities for retreat and the feeling of being surveilled.

Positive and negative feelings were also analyzed: in fact, living alone and in crowded dwellings [3] positively contributed to the positive feelings. Living in under-occupied dwellings [3] negatively impacted them.

Overcrowding negatively impacted on the negative feelings, in particular on aggression [63], and frustration [59]. Eventually, no statistically significant correlations were found between anxiety and living alone [3], nor between depression and constant or decreasing trajectory of household overcrowding over time [59]. Similarly, depression was never found to be correlated with a decrease in household density [59], nor stress with the presence of children under 10 in the households [77].

### 3.4. Neighborhood

This feature was examined in 13 studies. On average, most of the studies [3,6,60,61,66,72,73,77,79] referred to the type of neighborhood, classifying it as urban, semi-urban/suburban, or rural. The remaining studies analyzed the quality of the neighborhood, basing it on the poverty line and rates of unemployment [78], conditions of services and infrastructures [67], as well as social (such as percentage unemployed) and physical (such as number of abandoned buildings) attributes [65,66].

Regarding the IEQ parameters, associations were found between IAQ [67] and acoustic comfort [6,60,61,65]. Concerning the first parameter, the concentration levels of diesel particulates, an exposure present around homes, strictly depends on the area where the house is located. Thus, the more there is traffic in an area, the more their levels increase [67]. 

With regard to acoustic comfort, it was found to be higher when occupants were in natural environments and thus exposed to natural sounds [60]. Instead, negative associations were found with urban environment, where outdoor sources of noise (such as heavy traffic, ongoing construction, roads, trails, trains, aircrafts, parking garage, etc.) increased the level of noise annoyance [61]. When addressing the content of the environment i.e., the saturation of the environment with indoor and outdoor sounds or events, a higher content was registered when living in urban compared to suburban or rural areas [60]. No statistically significant correlations were found between acoustic comfort and urban areas when examined noise from sirens and industries [60].

Regarding mental health outcomes, associations were found with overall mental health [6,65], depression [6,77,78], anxiety [6,77], stress [6,65], and sleep problems [6,61]. Overall mental health was found worse when occupants were exposed to urban and traffic noise [6]. It was better when relocating from low-income neighborhoods to middle-income areas for both adults and children [65]. It also depends on neighborhood quality [65].

When examining depression outcomes, negative correlations were found when living in urban areas [77] compared with rural environments [6] and depending on the level of neighborhood socio-economic deprivation [78]. Two studies addressed the anxiety outcome, finding it to be worse when living in an urban area [77]. Not having a view of nature also increased anxiety symptoms [6].

Higher levels of stress were correlated with living in neighborhoods close to airport [6,65]. This had detrimental effects, especially on children aged 8–11 when exposed to chronic aircraft noise. Additionally, living in urban high-density areas [6] concurred with increased stress levels because of the limitations of social interactions. Our understanding of sleep problems was deepened by Andargie et al. [61] when they observed an increase for those living in urban areas due to exposed to noise from traffic.

Finally, no statistically significant correlations were found between overall mental health and urbanicity, the urban compared to rural environments [3], and the neighborhood quality [66]. Similarly, anxiety, loneliness, and positive feelings were never found to be related to living in natural or rural environments [3].

### 3.5. Green Spaces

This feature was examined in eight studies. Some of them referred to the presence of green elements on a residential property [2,3] such as a private garden, balcony, terrace, yard, shared gardens, green roofs, and walls. Other studies referred to their presence in the neighborhood [6,7] via places such as gardens, parks, pathways, playgrounds, and other greenspaces.

In general, on average, most of the studies referred to the view of nature through windows [2,6,60,65,71] or recreated in pictures or paintings [6,65]. Finally, four studies [2,6,65,66] specifically referred to a physical interaction with nature such as walking in the woods, reading in a garden, gardening, and growing plants in the home. 

Regarding the IEQ parameters, associations were found only with acoustic comfort [7,60]. It was found to be higher when working in a room facing a quiet area [60], and when exposed to natural sounds [7,60] and thus being in the presence of nature. 

Associations about mental health outcomes were found with overall mental health [2,3,6,60,65], depression [2,3,6,60,65], anxiety [2,3,6], stress [2,6,7,60], cognitive fatigue [2,6,60,65], positive feelings [2,6], negative feelings [2], and productivity [6].

Concerning overall mental health, the studies highlighted that it was worse when lacking access to outdoor spaces [3]. It was better when occupants could physically touch plants and soil [6] and view nature [6,60,65], landscapes painting, and indoor plants [65].

Being exposed to natural environments [6] and being in the presence of green elements that provided greater opportunities to practice physical and leisure/recreational activities also contributed to better mental health. Other predictors for an improved mental health were spending 20–25 min in the natural environments for those aged 30–60, and gardening [2].

When addressing depression outcomes, negative effects were found with poor-quality views from the windows [6,60] and lack of private gardens due to low-income housing areas [65].

Positive effects were found with access to outdoor spaces [3], the presence of green elements that provided greater opportunities to practice physical and leisure/recreational activities, and gardening [2].

Across the three studies assessing anxiety outcomes, all reported positive impacts associated with having access to outdoor spaces [3], viewing nature compared to viewing an urban area [6], and the presence of green elements that provided greater opportunities to practice physical and leisure/recreational activities [2]. Lower levels of stress were correlated with the presence of windows with access to nature eliciting positive emotions and stimulating recovery [60], access to nature [7], sweeping vistas and views of nature [2,7], and gardening [2,6].

Five studies found positive correlations between a lower cognitive fatigue and the presence of windows with access to nature eliciting positive emotions and stimulating recovery [60], exposure to nature or natural environments [6,65], contact with nature [6], and gardening [2]. Two studies deepened the relationship between an increase in positive feelings (such as improved self-esteem), touching plants and soil [6], and gardening [2]. With regard to negative feelings, gardening was found to produce a positive impact on anger. The presence of terraces and gardens were found to be responsible for a decrease in aggressivity [2]. 

The last outcome, productivity, was examined in one study [6] that showed the positive impact of viewing nature on it. Finally, no statistically significant correlations were found between having access to outdoor spaces, loneliness and positive feelings [3].

## 4. Further Correlations

Some mental health outcomes also depended on other factors beyond those already examined in the studies. Therefore, we decided to analyze even the relationship between them, and other information provided by the selected articles. Hence, the three groups were identified as follows:4.1IEQ parameters4.2Social features4.3Additional housing features

### 4.1. IEQ Parameters

This group referred to the impacts caused by IEQ parameters on mental health outcomes but did not considering housing conditions.

#### 4.1.1. Indoor Air Quality

Overall mental health was worse when carbon dioxide was above certain concentrations, as Engineer et al. [7] stated. Additionally, it was worse when there was long exposure to air pollutants [6,23], airborne and behavioral toxins [6,65], organic solvents [6], outdoor ambient pollutants and malodorous pollutants [65], and indoor air chemicals [80]. 

When higher concentration levels of volatile organic compounds (VOCs) were present, the overall mental health also got worst due to the production of sick building syndrome (SBS) [80]. Instead, it was better with excellent levels of IAQ.

When addressing depression outcomes, negative correlations were found with long-term exposure to air pollutants [23,67], nitrogen dioxide [67], mercury manganese organic solvents [65], and fine particulate matter (specifically, 2.5 nanometers or smaller PM_2_._5_ which may contain endotoxins) [6,67].

Negative impacts on anxiety were produced by exposure to fine particulate matter (specifically, 2.5 nanometers or smaller PM_2_._5_ which may contain endotoxins) [6], chronic exposure to airborne organic solvents [6], and exposure to mercury manganese organic solvents [65].

Regarding stress, lower levels were correlated with the presence of natural fragrances [7], while higher levels were associated with long-term exposure to air pollutants [23]. The cognitive fatigue outcomes were negatively linked with CO_2_ levels above approximately 950 ppm (by impairing cognitive performance by 15%) and above 1400 ppm (by reducing it of the 50% with fatigue and poor judgment also set in) [7].

Another predictor of higher levels of cognitive fatigue was long-term exposure to air pollutants [23]. Negative feelings of irritability were higher when exposed to pollutants [65], while aggression and frustrations were negatively influenced by the presence of toxins and pollutants [65].

Finally, no statistically significant correlations were found among particulate matter [67] and airborne VOCs [80] and overall quality of life [67].

#### 4.1.2. Thermal Comfort

When stress was examined, lower levels were associated with comfortable temperature and humidity [7], while higher levels were associated with thermal discomfort [7,72,79].

Assessing sleep problems, they were found to increase when dysregulations of thermal comfort occurred by affecting sleep time, sleep state maintenance, REM cycle length, and sleep efficiency [7].

Lastly, cognitive fatigue was negatively influenced by thermal discomfort [7].

#### 4.1.3. Lighting Comfort 

The overall mental health was worse due to a disturbance of the circadian rhythms [64,81] and the lack of natural daylight exposure [82]. Positive impacts were instead produced by circadian alignments that provide the optimal light conditions that impact circadian phase, sleep, and daytime energy levels [64]. 

Exposure to natural light also contributed to better mental health [6]. Examining the depression outcomes, negative impacts were produced by circadian misalignment, irrespective of total time spent asleep [64]. The lack of natural daylight exposure [81,82] and a short-term exposure to night-time light [82] were also responsible for higher levels of depression symptoms.

Other predictors were irregular light schedules [82], deficiency of daylight or its spectral anomaly [81], and disturbances to the circadian rhythms [81]. Exposure to bright light and light therapy [82,83] had instead a positive impact on depression.

Anxiety levels were found to be higher because of the circadian misalignment [64] and lack of natural daylight exposure [82].

Sleep problems were decreased by following the local day–night cycle [64], light, and light therapy [82]. Additionally, decreases were registered when exposed to adequate night light levels by following the circadian cycles, producing serotonin [81].

Instead, sleep problems were found to be increased by a lack of natural daylight exposure [82], and circadian rhythms not respected by altering sleep–wake cycles [82].

Deficiency of daylight or its spectral anomaly also caused higher levels of sleep problems [81] due to the fact that light induces the melatonin secretion (the sleep hormone). The cognitive fatigue outcomes were examined as being negatively influenced by circadian misalignment [64] and a lack of natural daylight exposure [82]. Eventually, productivity was higher in the presence of dynamic light (lighting that varies in color and illuminance during the day) [6] and adequate visual comfort [71].

#### 4.1.4. Acoustic Comfort 

Overall mental health was worse due to hearing more sounds from people at home [60], and perceived dominance of TV sounds and music during relaxation [60]. It was also worsened by exposure to traffic noise [23] and acute noise [61,65].

Otherwise, better mental health was achieved thanks to comfortable indoor soundscapes, lower noise sensitivity, lower dominance of neighbors’ noise, and music and TV sounds while relaxing [60].

A positive correlation was found between depression and exposure to indoor noise pollution [60], as well as with stress and exposure to construction noise [61] and loud exterior noise sources [65].

Lower stress levels were instead associated with nature sounds and quiet music [7]. When assessing sleep problems, they were found to be increased by exposure to construction and neighborhood activities compared to indoor noise sources [61]. They were also elevated when noise exposure increased [84].

With regard to loneliness, lower levels were associated with outdoor sounds that create a connection with the outdoor environments [60]. Cognitive fatigue outcomes were decreased by natural sounds [7]. Higher negative feelings about irritability and aggression were correlated with noise exposure [6,65]. Lastly, higher levels of anger were associated with exposure to outdoor noise levels [61].

#### 4.1.5. Overcrowding

Overcrowding was examined and found to cause a worsening in overall mental health [2,8]. It was also correlated with higher level of depression [8], anxiety [2,8], and stress [7] and sleep problems [2,8]. Overcrowding could indeed lead to the limitation of the activities that household members can comfortably undertake in the home, also causing conflicting activities among the household members such as watching television and studying.

### 4.2. Social Features

This group included general information about the population of respondents such as their age, gender, and socio-economic status (such as being workers or unemployed). Information about the personal and housing precarity and the COVID-19 pandemic were also included.

#### 4.2.1. Population 

Worse mental health was observed more in female than male respondents [60]. Higher depression levels were registered amongst divorced and widowed persons [59], as well as in the female population and in people younger than 40 [77].

Anxiety was found to be higher in people aged 18–25 [1], in female population and in people younger than 40 [77]. Similarly, stress was found to be higher in people aged 18–25 [1]. Finally, sleep problems were found to increase with age in women [84], while in total, decreased in women that on average slept 54 min longer than men considering all times of the year [83].

#### 4.2.2. Precarity

Starting from several studies, overall mental health was worst in people experiencing double precarity (housing and employment) through job loss, job insecurity, and insecure tenancies [6,63]. Unemployed people [7], people in lower socio-economic groups [6,84], and people exposed to socio-behavioral and socio-economic disadvantages were also more susceptible to lower levels of mental health [67]. Additionally, adults that were exposed to poverty from birth to age 9 [73], people experiencing fuel poverty (being unable to adequately heat their home in winter), and people having debts [79,85] were found to have poorer mental health. Depression levels were higher because of debts, social status, and unemployment [59,78].

Living in low-income households [7,77] and experiencing housing instability [6] were other predictors for depression symptoms. Additionally, youth aged 14–21 were estimated to experience higher levels of depression due to poverty in early life [73], mortgage debts, and fuel poverty [79].

Anxiety levels were found to be higher in low-income households [77], and due to housing instability [6]. Youth aged 14–21 were found to experience higher levels of anxiety due to poverty in early life [73], mortgage debts, and fuel poverty [79]. When addressing stress outcomes, negative impacts were produced by debts [59] and unemployment [77].

In general, people who lost income and who lived in economic insecurity [77] were found to be more susceptible to higher levels of stress. Additionally, low-income households [7], and youth aged 14–21 were found to experience higher levels of stress due to poverty in early life [73], mortgage debts, and fuel poverty [79].

In conclusion, positive feelings were lower in unemployed people with regard to self-esteem [78]. Finally, when addressing motivation, it was found to be lower due to early childhood poverty [73].

#### 4.2.3. COVID-19 Pandemic

Overall mental health was worse+- in people exposed to COVID-19 shocks [63] and due to economic and social impacts of the pandemic [7]. The COVID-19 pandemic also contributed to an increase in depression, anxiety, and stress levels [7], especially in the first months of the pandemic [77]. Sleep problems and loneliness were also found to be increased due to COVID-19 [7].

Finally, the pandemic also undermined positive feelings about motivation [7], while enhancing negative ones about irritability and aggression [7,86,87,88].

### 4.3. Additional Housing Features

This group included two additional features related to the built environment but that are not directly IEQ parameters.

#### 4.3.1. Spatial Design

Intended as the design of interior environments, their layouts and characteristics such as adequate and flexible spaces. Positive correlations were found between mental health and design of the built environment that support and encourage social interactions [7]. 

Lower levels of stress were registered when the layout of spaces encouraged movement and promote physical activity [7]. On the contrary, inadequate spatial design increased the cognitive fatigue and reduced the productivity [7].

#### 4.3.2. Occupant Control

This is defined here as the capacity of people to control their surroundings and maintain a sense of self-efficacy via behaviors such as physically altering lighting, temperature levels, and furniture position. A lack of control over one’s environment, like the temperature of the space, was proven to cause a worsening in overall mental health, cognitive fatigue, and productivity [6].

Otherwise, better mental health [65], lower stress, and higher positive feelings about motivation [6] were registered when people could control their surroundings. 

## 5. Limitations and Further Developments

Some limitations to our results and external validity needed to be listed and taken into consideration:first, this was a scoping review which was limited to only one database (Scopus), leading to the possibility that some studies may have been missed for many reasons;secondly, we limited our search to articles published in English and from 2012 to June 2022. It should be noted that this topic is constantly updated, especially if we consider the publication of studies and research subsequent to those analyzed by us, whose publication took place by June 2022;thirdly, the exclusion of some typologies of contributions such as conference proceedings, dissertations, and grey literature was another limitation: therefore, it could be helpful to extend the review in order to include some selected relevant non-peer-reviewed studies;fourthly, the mental health outcomes addressed were grouped in macro-domains, such as depressive symptoms, anxiety symptoms, stress levels, and even elusive categories such as “*overall mental health, positive feelings and negative feelings*”.

The same heterogeneity permeated the chosen five housing conditions, which were grouped in macro-domains. Macro-domains were used due to a lack of lacking, making it impossible to practice in-depth investigations by breaking down each domain. However, this could be seen as a strength of the review; indeed, it is not restricted to only one housing characteristic and one mental health outcome. In fact, we established associations between several mental health outcomes and several housing conditions.

Another limitation was the methodological heterogeneity of the selected studies since the outcomes addressed were not always measured in the same way throughout the different studies. Future research should also compare the tools used to derive the outcomes addressed.

We also would like to point out that the main limitations to our study were related to the typology of data available. First, we did not find in the existing literature exhaustive data about quantitative and technical information. Indeed, correlations between the three domains (IEQ, Housing Conditions, Mental Health) were hard to identify. Thus, our collection of evidence from the literature was merely qualitative. More in-depth analyses are needed to quantify the measures for making our homes healthier. Second, associations with low–middle-income countries (LMIC) could not be drawn since they present a wide disparity both in housing conditions and mental health status compared to OECD country. Few studies [89,90,91,92] found in the literature addressing the topics of this paper in under-developed countries were not sufficient to be taken into account. Our results need to be tailored to a specific geographical context in order to establish more detailed associations. It could be useful to define a broader set for the discussion, so that a worldwide contribution will be provided.

Future lines of developments should fill these gaps we pointed out, by making more data available. This will improve comparability between data from different contexts, also reducing uncertainty.

Finally, it must be noticed that there are multiple gaps “*[…] non-housing variables that affect health (such as poverty, ignorance, poor nutrition, lack of medical care*)” [93] that are hard to separate or to take into account when addressing housing and health studies. Additionally, the cause-and-effect relationship between housing and mental health variables is often unclear. Future researchers could try to further develop these aspects. Due to their complexity, it may be useful to address one housing conditions at a time by linking it to multiple mental health outcomes.

## 6. Conclusions and Perspectives

Despite the above-mentioned limitations inherent to the current scoping review, we can state that a collection of data regarding the impact of Housing Conditions on Mental Health has been conducted. Specifically, we identified in the existing literature five housing conditions [house type, age, and floor level; housing qualities; household composition; neighborhood; green spaces] that affect the mental health outcomes of inhabitants in a major way. 

This review paper is therefore intended to be the first reference point in providing a cognitive overview of the qualitative data available. These will serve for future research, by suggesting which factors need to be most and well addressed in order to make our living settings healthier, more resilient, and salutogenic.

Starting from the position paper by Signorelli et al. [94], the qualitative design suggestions identified in the selected studies can be summarized as follows:optimizing IAQ by limiting the use of construction materials containing toxins or emitting VOCs [6], and through adequate ventilation systems [7];increasing outdoor air ventilation via integration with design that reduces energy consumption [6,76];optimizing daylight through a current orientation [2], and by ensuring the proper levels, timing, duration, and spectra of light that occupants are exposed to [64];creating a healthy and supportive environment by improving the acoustic performance [6,60,61];solving overcrowding [59];solving the housing deficit [59], and promoting a design that encourages physical activity, and spaces to congregate and for quiet contemplation in the house [7];creating home workspaces in small living quarters [7];introducing a biophilic design that fosters a positive human–environment connection [2,6,7].

Additionally, qualitative socio-economic suggestions were found in the selected studies to better address the mental health of people in residential spaces, such as:implementing psychosocial programs [77], employment assistance service [10], quality education opportunity [10], and community-based support mechanism [3];fostering tax incentives for building renewal [2], and funding for affordable housing programs [8,59].

It is clever that these suggestions are far from being exhaustive best practices to reach more resilient houses. Although these recommendations are coherent with the WHO Guidelines for Healthy Housing indeed, the limitations mentioned above make it impossible to provide more detailed recommendations. As WHO stated, “*[…] the lack of detailed epidemiological information relating to conditions in developing countries and the wide disparities in geography, culture, social habits and political priorities means that these guidelines are inevitably very generalized*” [93]. Future research could undertake a comparison between the evidence we collected in this paper and the WHO guidelines. 

Finally, we wanted to highlight how the present review has systematized the knowledge and evidence available on a topic that is currently more debated and subject to continuous updates. Therefore, this paper is to be considered as an open contribution that can be implemented both starting from the inclusion of further areas not explored here, and from the integration of more recent studies that can contribute to the updating and deepening of the work here presented.

## Figures and Tables

**Figure 1 ijerph-19-15975-f001:**
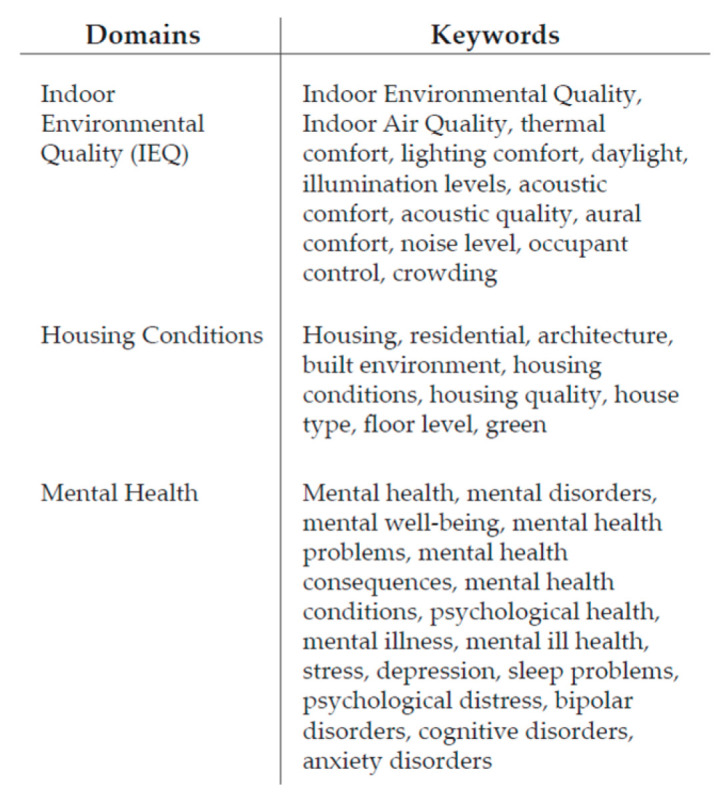
Search settings.

**Figure 2 ijerph-19-15975-f002:**
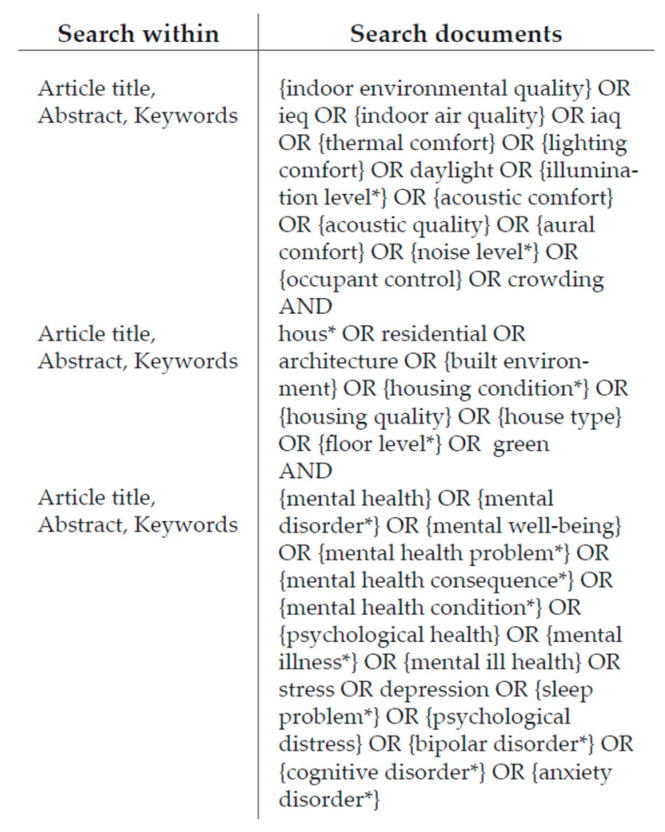
Search strings.

**Figure 3 ijerph-19-15975-f003:**
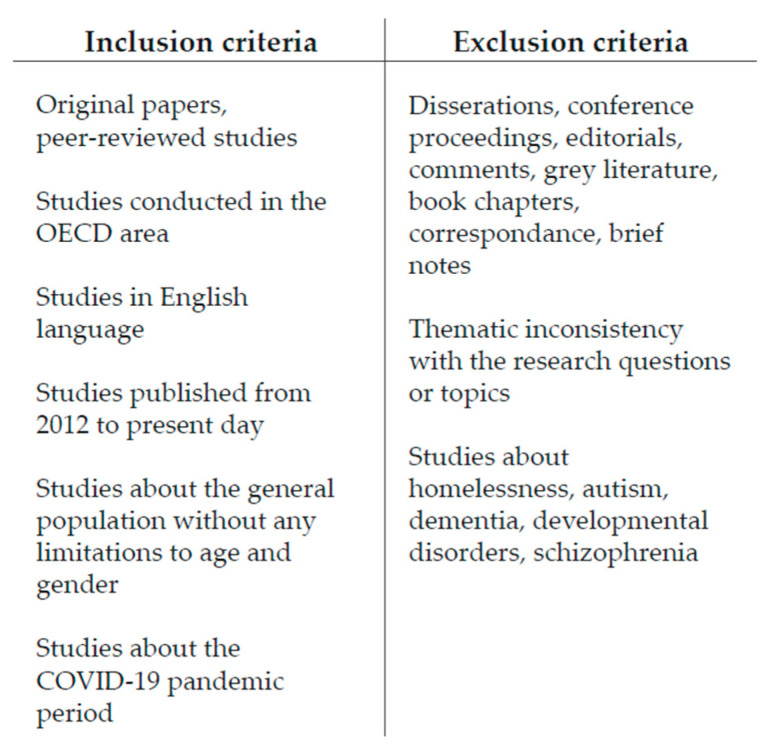
Eligibility criteria.

**Figure 4 ijerph-19-15975-f004:**
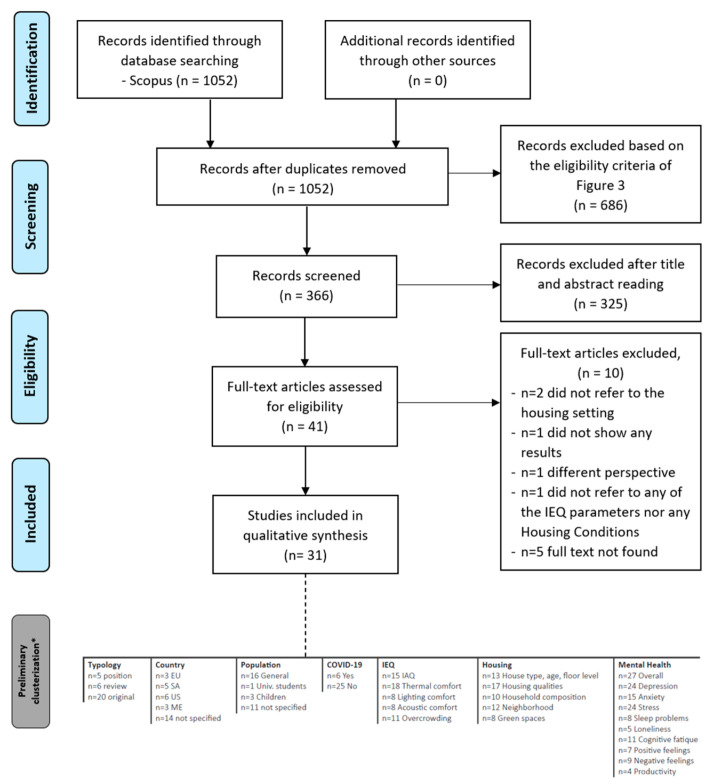
PRISMA flowchart diagram.* Described in Section 2.6.

**Figure 5 ijerph-19-15975-f005:**
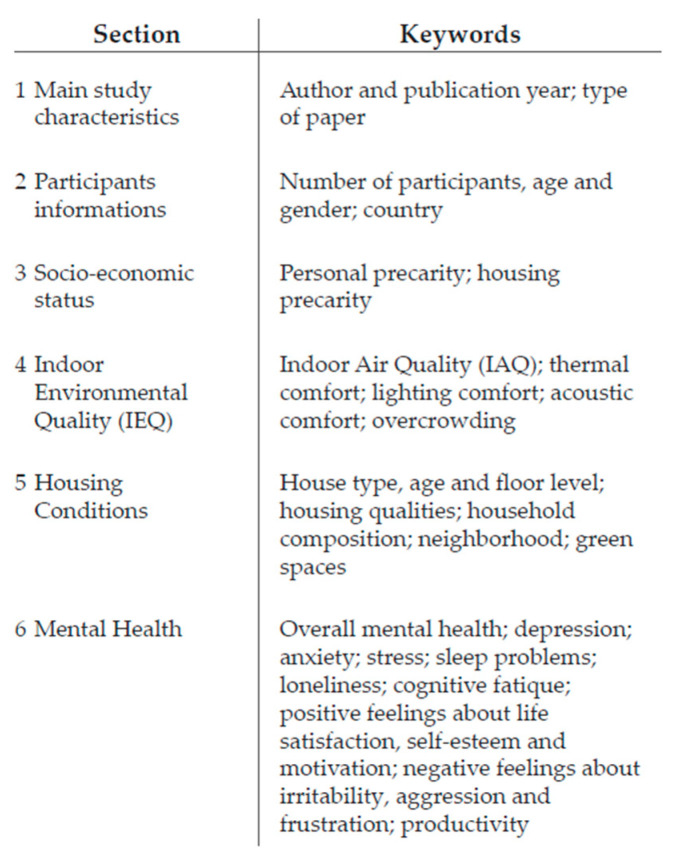
Clusterization of sections.

**Figure 6 ijerph-19-15975-f006:**
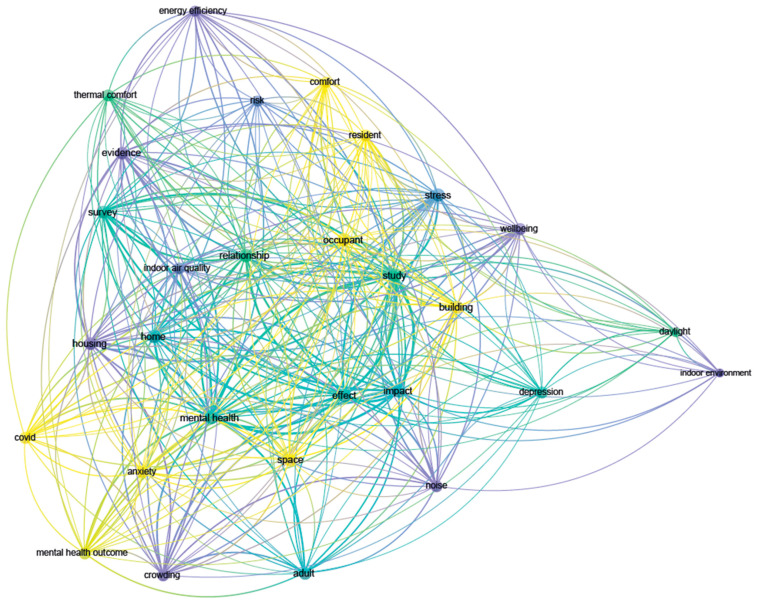
Overlay visualization: average publication year from VOSViewer. The colors indicate the average publication year: purple (2018), green (2019), yellow (2020).

**Figure 7 ijerph-19-15975-f007:**
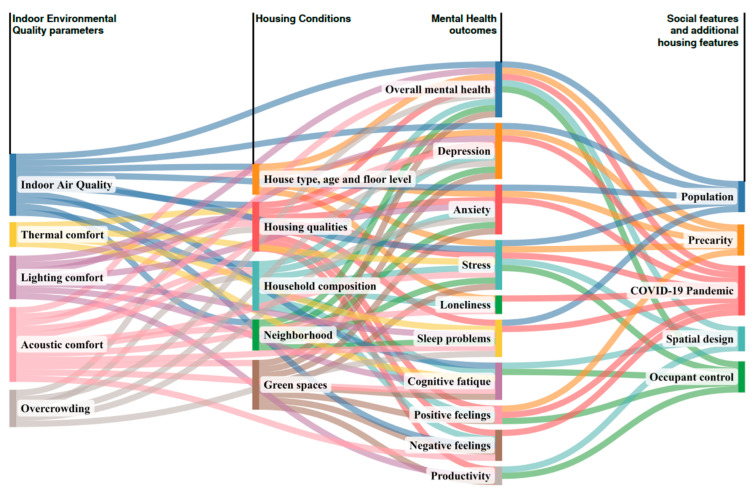
Correlation among IEQ, housing conditions, mental health outcomes and social features, and additional housing features.

## Data Availability

Not applicable.

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
