# Peer review of "Can Homes Affect Well-Being? A Scoping Review among Housing Conditions, Indoor Environmental Quality, and Mental Health Outcomes"

_ijerph, 2022, doi:10.3390/ijerph192315975_

Round 1

Reviewer 1 Report

The paper explores the relationship between Housing Conditions, Indoor Environmental Quality (IEQ) and Mental Health implications on human well-being. The research topic is interesting and significant. However, the article has some significant shortcomings that can be not ignored. 

1.     The paper seems merely collect all existing literature, lacking in-depth analysis. Many viewpoints are those that people already reach a consensus on. For example, “noise annoyance was higher in older than newer buildings and among respondents in lower floors” (see line 408); “overall mental health was found to be better when having more windows in a room by guaranteeing the right amount of daylight exposure and a pleasant lighting quality having adequate space in the home, after housing improvements such as new kitchens, bathrooms, and electrics, and after household energy efficiency interventions also by reducing household energy expenditures” (see line 473-478).

2.     The authors point out, “Although many studies established a relationship between the built environment and its effects on physical health, its impact on mental health and well-being, it is a very recent issue, with a growing body evidence that still lacking quantitative data and that does not consider the multiple factors by which the built environment is composed” (see line 66-69). Thus, it is suggested that the authors analyze and summarize quantitative data and/or other more valuable findings (e.g., the relationship among multiple factors) related to the research topic, rather than list what other scholars did. In this way, the paper may also address the shortcoming (see aforementioned point 1).

3.     The argument, “a recognition of design recommendations and best practices that will help to reduce inequalities in housing, by making them healthier, more resilient and salutogenic” (see the abstract, lines 27-29), is far from enough. I would suggest the authors rewrite the last section of the paper.

4.     Many sentences are too long to read and understand, e.g., lines 473-478, line 586-590, etc.

5.     Some sentences are incomplete, see lines 362, 491, and 498.

6.     Lines 314-346 are suggested to be synthesized as a chart.

Author Response

  1. We are fully aware of that, since our purpose was precisely to synthetize the existing literature and provide an exhaustive collection that will serve as a starting point for future research. Following yours and others reviewer suggestions, we tried to better clarify this aspect both in the Introduction (see lines 70-76) and in the Conclusion paragraph (see lines 933-934).
  2. We are fully aware of that, and following your suggestion we tried to better clarify this limitations of our study, as we stated in the Introduction (see lines 65-67) and in the Limitations and further developments (see lines 903-908) paragraphs. The lacking of quantitative data was well-observed during the collection of evidence from the existing literature, making it impossible to provide technical informations that went beyond the qualitative data we collected. The huge gap between qualitative and quantitative data regarding these topics should be filled by future researchers.

  3. According to your suggestion, we rewrote the Conclusion and perspectives paragraph. 

  4. Following your advice, we re-edit many parts of the manuscripts to enhance its readability and comprehension.

  5. Following your advice, we re-edit many parts of the manuscripts to enhance its readability and comprehension.

  6. Lines 314-346 (now lines 336-368) are already the explanation of the chart reported in the final part of Figure 4 at Page 9 (PRISMA flowchart diagram-preliminary clusterization), as lines 334-335 stated “From the grid developed by the authors, the following preliminary clusterization emerged, as reported in the final part of Figure 4, the 31 studies analyzed have been sub-divided into:

Reviewer 2 Report

The paper deals with a relevant topic, the approach used is appropriate and the scoping review is well developed. However, there is a difference in quality between the discussion of the results of the scoping review, which are well described, and the development of the topic relating to best practices, which is dealt with only in the concluding paragraph. The paragraph dedicated to Conclusions and Perspectives is in fact absolutely insufficient and detracts from the rest of the work. The complexity of the question is resolved by reporting in a very generic and not very useful way, what the WHO housing and health guidelines (which the same authors cite) deepen in a very comprehensive report. The authors, for example, could clarify whether the articles analyzed contain indications regarding possible solutions and practices and possibly compare them with the best practices suggested by the WHO report. If this is not feasible, the authors could enrich the analysis by further developing the lacking aspects that emerge from the scoping review. However, the debate on the possible lines of development of research on the topic discussed by the authors must also be reviewed and enriched.

Author Response

According to your suggestions, we rewrote the Conclusion and perspectives paragraph, trying to better clarify the purpose of our paper. 

We reported some qualitative suggestions emerged from the selected studies. A comparison between them and the WHO guidelines though, is far from our purpose. This could be a future line of development of our research, as we  stated in lines 968-969. The complexity and the vastness of such a theme will need a proper paper. Nevertheless, as stated by WHO itself and reported in lines 965-969, their guidelines are also qualitative and generalize.

According to your suggestions, we wrote a new paragraph (Limitations and further developments) that pointed out the lacking aspects of our study, as well as possible new lines of development of our research.

Reviewer 3 Report

Language: There is a need for the authors to re-edit the entire manuscript to enhance its readability and comprehension. One major issue the authors must address are the too long, complicated, and convoluted sentences that all over the manuscripts. Writing the manuscript in short and simple sentences will enhance the quality of the manuscript. 

Statement of Gap: There is a need to strengthen gap that the review paper claim to be responding to. The gap the manuscript sought to address was not visible and clearly delineated, neither were the contributions properly scoped. The manuscript must address these major concerns.

Conclusion: The conclusion needs to be rewritten by including the implications of the findings for practice and policies aimed at fostering resilient and healthy homes. Similarly, the conclusions should highlight the implications of the findings for future studies by suggesting potential  research areas for exploration. Finally, the conclusion should highlight the limitations of the study and the implications for their findings. 

Author Response

Language: Following your advice, we re-edit many parts of the manuscripts to enhance its readability and comprehension.

Statement of the Gap: According to your suggestion, we tried to better clarify the purpose of our paper (see lines 70-76, 933-936).

Conclusion: According to your suggestions, we rewrote the Conclusion and perspectives paragraph, trying to better clarify the purpose of our paper. Also, we wrote a new paragraph (Limitations and further developments) that pointed out the lacking aspects of our study, as well as new lines of possible developments of our research.

Reviewer 4 Report

Summary:

This research article is investigating the relationship between Housing Conditions, Indoor Environmental Quality (IEQ) and Mental Health implications on human well-being. Due to Covid, people are forced to stay at home and deal with many unprecedented conditions. Mental health issues are becoming more and more important and should be equally addressed compared to physical health issues. The scope of this study is broad but well-defined by identifying the keywords related to the studied topics. I think this review article is worth publishing, and future studies should focus on these important topics.  

Major issue:

This paper does have one limitation. In my opinion, the authors may need to set a context for the discussion. The topics of this article such as green space are more important in developed urban areas. I feel like the authors are missing the low-middle-income countries. Should future studies focus on these areas? I feel like the importance of housing conditions and indoor environmental quality are hugely different between developed countries and LMIC (low-middle-income countries). In developed countries such as the USA, CDC has some information on mental health prevalence in the population. Also, housing and indoor environments are more defined in developed countries. When we look at LMIC countries, both housing conditions and mental health status are very limited studied. Therefore, I think authors need to have some texts to address the huge knowledge gaps among areas in the world. We should not only focus on the developed areas which have a smaller amount of population compared to the rest of the world. 

Author Response

We are aware of that, but as we stated in lines 241-246 we focused our attention on developed countries of the OECD area. Thus, green spaces were considered as Housing Conditions because of their impact in this context.

Yes, future studies should focus on low-middle-income countries. According to your suggestion, we better clarify the need for future research among the LMIC (see lines 908-915).

According to your suggestion we reported some articles regarding the huge gaps among areas in the world (see footnote n.2).

Round 2

Reviewer 2 Report

All the points indicated and suggested have been satisfactorily addressed by the authors

Author Response

Thank you again for your valuable suggestions.